# Nutritional Status Disorders and Selected Risk Factors of Ventilator-Associated Pneumonia (VAP) in Patients Treated in the Intensive Care Ward—A Retrospective Study

**DOI:** 10.3390/ijerph19010602

**Published:** 2022-01-05

**Authors:** Lucyna Ścisło, Elżbieta Walewska, Iwona Bodys-Cupak, Agnieszka Gniadek, Maria Kózka

**Affiliations:** 1Department of Clinical Nursing, Institute of Nursing and Midwifery, Faculty of Health Sciences, Jagiellonian University Medical College, 31-501 Krakow, Poland; lucyna.scislo@uj.edu.pl (L.Ś.); elzbieta.walewska@uj.edu.pl (E.W.); maria.kozka@uj.edu.pl (M.K.); 2Laboratory of Theory and Fundamentals of Nursing, Institute of Nursing and Midwifery, Faculty of Health Sciences, Jagiellonian University Medical College, 31-126 Krakow, Poland; 3Departement of Nursing Management and Epidemiology Nursing, Institute of Nursing and Midwifery, Faculty of Health Sciences, Jagiellonian University Medical College, 31-501 Krakow, Poland; agnieszka.gniadek@uj.edu.pl

**Keywords:** VAP, ventilator-associated pneumonia, risk factors, nutritional status disorders, intensive care

## Abstract

Introduction: The development of pneumonia in patients treated in intensive care wards is influenced by numerous factors resulting from the primary health condition and co-morbidities. The aim of this study is the determination of the correlation between nutritional status disorders and selected risk factors (type of injury, epidemiological factors, mortality risk, inflammation parameters, age, and gender) and the time of pneumonia occurrence in patients mechanically ventilated in intensive care wards. Material and method: The study included 121 patients with injuries treated in the intensive care ward who had been diagnosed with pneumonia related to mechanical ventilation. The data were collected using the method of retrospective analysis of patients’ medical records available in the electronic system. Results: Ventilator-associated pneumonia (VAP) occurred more frequently in patients over 61 years of age (40.4%), men (67.8%), after multiple-organ injury (45.5%), and those with a lower albumin level (86%), higher CRP values (83.5%), and leukocytes (68.6%). The risk of under-nutrition assessed with the NRS-2002 system was confirmed in the whole study group. The statistical analysis demonstrated a correlation between the leukocytes level (*p* = 0.012) and epidemiological factors (*p* = 0.035) and the VAP contraction time. Patients infected with Staphylococcus aureus had 4% of odds for the development of late VAP in comparison to Acinetobacter baumannii (*p* < 0.001), whereas patients infected by any other bacteria or fungi had about four times lower odds of the development of late VAP in comparison to Acinetobacter baumannii (*p* = 0.02). Patients with results in APACHE from 20 to 24 and from 25 to 29 had 13% and 21%, respectively, odds of the development of late VAP in comparison to patients with APACHE II scores ranging from 10 to 19 (respectively, *p* = 0.006; *p* = 0.028). Conclusions: The development of VAP is impacted by many factors, the monitoring of which has to be included in prophylactics and treatment.

## 1. Introduction

Ventilator-associated pneumonia (VAP) is defined as an infection of lung parenchyma in patients subjected to invasive mechanical ventilation for at least 48 h, and is a part of pneumonia acquired in intensive care units (ICU) [1,2,3]. Infections to the respiratory system, including VAP, occur in 16% of patients treated in intensive care wards [3,4]. Its mortality rate is at the level of 10%, and in the case of patients with pneumonia in whom the etiological factor is multi-resistant micro-organisms, it reaches 40% [5,6,7]. VAP remains one of the most frequent infections in patients which requires invasive mechanical ventilation, thus posing an economic load on the healthcare system. Despite the progress in the area of microbiological research, epidemiology, and diagnostic criteria, many issues related to VAP still remain controversial within prophylactics and treatment [8].

The data published by the Infectious Diseases Society of America (IDSA) and American Thoracic Society (ATS) demonstrate that VAP-related mortality in the USA reaches 13% [9,10]. Though in Europe a multicenter prospective study showed that 30-day mortality due to VAP was 29.9%; early-onset VAP mortality was 19.2%, and late-onset VAP mortality reached 31.4% [11].

The risk of VAP occurrence is the highest in the early period of hospitalization, and increases in the case of bacterial flora translocation from the oral cavity, causing lower respiratory tract infections as a result. In a vast majority of cases, VAP etiology is monobacterial [6]. The most common reasons are the following bacteria: *Streptococcus pneumoniae*, *Streptococcus pyogenes*, *Staphylococcus aureus*, *Pseudomonas aeruginosa*, *Klebsiella pneumoniae*, *Acinetobacter baumannii*, and *methycyllin resistant* (*methycyllin resistant Staphyllococcus aureus—MRSA*) strains, as well as *RSV* (*respiratory syncytial virus*) viruses, *human influenza A and B viruses*, *rhiniviruses*, *coronaviruses*, and *adenoviruses* [11,12,13]. Prolonged ventilation, moreover, increases the risk of infection caused by humidifiers and volume loops of the ventilator, which are the source of the pathogen due to exposure [14]. Other risk factors include: disorders of consciousness [15,16,17]; head injuries [18]; sedation delirium [19]; and underlying diseases, including coronary heart disease, diabetes, diseases of the respiratory system, chronic renal failure, and thyroiditis (Hashimoto disease) [16,20,21,22]. Also, some medical procedures fall into this group, such as reintubation, tracheostomy, and fiberoptic bronchoscopy [23]. Crucial factors also include: the patient’s weakened resistance mechanisms because the antibacterial function of the leukocytes and cellular immunological response are weakened, as well as the compromise of the cutaneous-mucosal barrier, the rapid increase of colonization, and the replacement of the physiological flora with the strains from the hospital. Furthermore, the trauma, or surgery itself, may cause disturbances to the immune system. Currently, researchers are trying to find reliable diagnostic markers, which include C-reactive protein (CRP) and procalcitonin (PCT), that might quickly and easily differentiate VAP patients [24]. Other risk factors of lung infections related to the patient are: multiple diseases, multi-drug therapy, chronic lung diseases, advanced age, under-nutrition, obesity, consciousness, and resistance disorders [6,25].

Based on the conducted research, the authors demonstrated that, all over the world, under-nutrition already occurs in patients at the time of admission to the hospital, and it also appears during the hospitalization, and in some patients, especially those treated in the ICU, it increases during the therapy. The observations of the decrease of the body mass index (BMI) in over 10% of patients, and serum albumin concentration in 20% of patients demonstrate that qualitative under-nutrition is the reason of the body mass decrease, and the lymphocyte and hemoglobin count drop, and, as a result, the increased risk of complications and a prolonged hospital stay [26,27].

It has to be mentioned that early-onset VAP, i.e., until the 4th day of the patient’s stay in the ICU, according to the definition accepted by the American Thoracic Society (ATS), is more frequently caused by external bacteria sensitive to antibiotics, which is related to a better prognosis. Late-onset VAP, however, i.e., from the 5th day after admission to the ward, is usually caused by multi-resistant micro-organisms from the hospital flora, which is connected to a worse prognosis. Fungi *Aspergillus*, as well as anaerobic bacillus of *Legionella pneumophila*, are the potential etiological factors of late-onset VAP due to the antibiotic therapy used [6].

In the last two decades, many centers have conducted studies related to the occurrence of VAP [28], which led to the implementation of prophylactic packages, which helped decrease infections, mortality, and cost of medical care [9,10,29].

There are a few strategies to prevent and control VAP, namely prophylactic packages and medications, including chlorhexidine, β-lactam antibiotics, and probiotics [30]. Although the frequency of VAP occurrence has dropped in the recent years thanks to the implementation of therapeutic strategies, it still remains one of the most frequent causes of hospital infections and death of patients in critical conditions while hospitalized in ICUs. VAP may prevent the patient from being disconnected from the ventilator, which enforces longer hospitalization, and increases financial load for the healthcare system and the demand for medical supplies [31]. Thus, it is essential to recognize VAP risk factors, while, at the same time, providing better prophylactics and control [32]. It is also important to undertake effective prophylactic actions in cooperation with the therapeutic team, as well as the instant recognition of VAP, and the implementation of intensive treatment.

## 2. Materials and Methods

### 2.1. The Aim of the Study

The aim of the study was to determine the correlation between selected risk factors (nutritional status at the time of admission into the ICU, type of injury, epidemiological factors, risk of death, inflammation parameters, age, and gender) and the time of pneumonia occurrence in patients with injuries who were mechanically ventilated in intensive care units.

**Hypothesis** **1.**
*The more risk factors, the higher probability of VAP development.*


### 2.2. Organisation and Course of the Study

The study was conducted in the Anaesthesiology and Intensive Care Clinical Department No. 1 in the Trauma Centre of the Department of Disaster and Emergency Medicine of the University Hospital in Cracow. The research included patients hospitalized in the period of time from 1 January 2017 to 1 January 2018.

The study was conducted in accordance with the principles of Good Scientific Practice, the rules of the Declaration of Helsinki, the Regulation (EU) 2016/679 of the European Parliament and of the Council of 27 April 2016 on the protection of natural persons with regard to the processing of personal data and on the free movement of such data, and repealing Directive 95/46/EC (General Data Protection Regulation), the Act of 10 May 2018 on personal data protection (Polish: Dz.U. 2018 poz. 1000—Journal of Laws from 2018, item 1000) [33,34]. The permission to conduct the study and use the data from the patients’ medical records was granted by the head of the department and the Hospital Director.

### 2.3. The Study Group

The study included 121 patients in medical intensive care units (MICU) with recognized injuries (multi-organ injury, craniocerebral trauma, post-surgery trauma, isolated spinal/pelvic injury) and diagnosed ventilator-associated pneumonia (VAP).

The inclusion criterion of the study consisted of mechanically ventilated ICU patients in whom the symptoms of pneumonia developed after at least 48 h from the intubation. In the study, the group of patients with multi-organ injury consisted of persons who had been in an accident/injured causing major injuries in at least two organs, which posed a risk of impaired cardiopulmonary stability in a patient, with each of those injuries requiring a specialistic treatment in the ICU due to the life-threatening condition.

The exclusion criteria from the study were patients mechanically ventilated while hospitalized in another ward with diagnosed pneumonia at the time of admission, and patients with symptoms of liver or renal failure.

Two groups of patients were distinguished on the basis of the time of VAP occurrence. The first group consisted of patients in whom VAP was detected in the early period within the first 4 days of mechanical ventilation application (early-onset VAP); the second group consisted of patients in whom VAP was recognized at a later time, at least 4 days from starting alternative breathing (late-onset VAP).

### 2.4. Methods, Techniques, and Research Tools

The research used the method of retrospective analysis of patients’ medical records, using daily observation and laboratory test results, and microbiological cultures from upper respiratory tract excretion.

In order to identify pathogens, the material was acquired from the upper respiratory tract using a non-invasive method (by tracheal aspiration) before the inclusion of antibiotics, according to the accepted procedure enforced in the clinical ward of the University Hospital. The obtained material was directly transported to the laboratory in order to obtain microbiological diagnostics.

To analyze the nutritional status, the results of the nutritional status assessment based on the NRS-2002 (Nutritional Risk Screening 2002) scale, developed in 2002 by a group of experts from the European Society for Clinical Nutrition and Metabolism (ESPEN), were used. The questionnaire is based on the results of randomized clinical trials, and the assessment is performed in a few stages, covering: risk of under-nutrition, the severity scale of the disease, and the initial and final screening. The risk of under-nutrition is assessed on the current nutritional status and anticipated risk of worsening of the current nutritional status due to the increased demand in the course of the disease, and the resulting increase of energy demand or caused by the surgery. A total of the obtained points > or = 3 means the risk of under-nutrition and the urge to start nutritional support (treatment) [35].

In all patients, at the time of admission, the risk of malnutrition was determined based on the NRS 2002 scale. According to the ESPEN recommendations, due to the patients’ clinical conditions, and taking into consideration the results of nutritional status assessment results in the NRS 2002 scale, parenteral nutrition was introduced. Nutrition treatment is prepared according to the clinical condition of patients and their metabolic needs. In the clinical ward where the study was performed, in order to determine caloric demands, a formula was implemented, according to which the aim of nutritional treatment was to provide between 20 and 25 kcal/kg/day for the patient. Non-protein energy from carbohydrates and fats constituted 50–65% of energetic demand. The supply of protein was 1.5–2.5 g/kg/day, and the demand for water calculated per kilogram of body mass was 30–40 mL. In our own studies, to assess the nutritional status of an adult patient admitted into the ICU, the NRS 2002 scale was used, which is recommended by the Polish Ministry of Health (Minister of Health regulation of 1 January 2012 as amended).

The study also used the APACHE II (Acute Physiology and Chronic Health Evaluation II) scale, which serves to assess the severity of the conditions of adult patients admitted into intensive care units, and to forecast the mortality risk. The scale allows the assessment of the condition of a patient based on their overview and physiological variables. The scale includes vital parameters, such as: rectal temperature, average blood pressure, heart function, respiratory parameters of mechanically ventilated patients, pH of arterial blood, sodium and potassium ion concentration, creatinine concentration, hemoglobin level, leukocyte count, and the result of assessment in the GCS (Glasgow Coma Score) scale [36].

A database was created using the Microsoft Office Excel calculation sheet containing the information on age, gender, injury mechanism, nutritional status of patients at the time of admission into the ward (total protein and albumin level in blood serum, the number of points in the NRS-2002 scale), the data on inflammatory indicators at the time of admission into the ward (acute phase protein, leukocytes, procalcitonin), the number of points in the APACHE II scale at the time of admission, the day of pneumonia diagnosis, and the epidemiological factor that triggered the inflammation.

To conduct the study, the following correct values of parameters indicating the patients’ nutritional status according to the reference norms of the laboratory performing the tests—University Hospital in Krakow—were accepted. Total protein in blood serum was accepted at 45–80 g/dL, and values below 45 g/dL are incorrect. Albumin in blood serum over 35 g/L, NRS-2002, was below 3 points. The albumin values included as a prognostic-nutritional indicator between 31 g/L and 35 g/L point to mild under-nutrition, values between 25 g/L and 30.99 g/L to moderate under-nutrition, whereas those below 25 g/L may indicate severe under-nutrition and the wrong prognostic factor. The following correct values of inflammatory indicators were accepted: acute phase protein (CRP)—below 5 mg/L; leukocytes count (WBC)—4000–10,000/µL; procalcitonin—below 0.5 µg/L [35].

### 2.5. Statistical Analysis

The collected materials were described using statistical analysis. The obtained information served to create a database in Microsoft Office Excel 2007 (Microsoft Corporation, Albuquerque, NM, USA). To perform statistical calculations, Statistica 10 software (StatSoft, Tulsa, OK, USA) for statistical calculations was used. The values of qualitative variables were presented by means of absolute values and percentage rate. For such variables, the significance of differences within particular groups was verified by the chi^2^ (χ^2^ Pearson’s) test. The values of quantitative variables were presented using mean values. The dependency of quantitative variables within particular groups for variables with normal distribution was tested with the parametric *t*-Student test for independent groups. For variables with distribution differing from normal distribution, the differences between the two independent groups were studied using a non-parametric U Mann–Whitney test. For the purpose of computation of all results, statistical significance at the coefficient level of *p* < 0.05 was assumed.

Assessment of the factors which influenced the development of the later VAP was conducted using the logistic regression model. The model was developed by starting from the null model, and by adding the independent variables one by one, starting from those with the most significant bivariate relationship with the dependent variable. At each step, independent variables which lost their significance were removed from the model. As independent variables, all analyzed variables were tested, but, for some of them, categories with low frequencies were aggregated: for total protein in blood serum—two higher categories; for WBC—two lowest categories; for APCHE II—two lowest and two highest, respectively, and a patient with spine injury was removed. In the final model, only variables which remained significant at the last stage of the procedure were retained. The results of the model were presented as odds ratios (OR), 95% confidence intervals (95% CI), and respective *p* values. The analysis was conducted using IBM SPSS Statistics v. 27 for Windows (IBM Corp, released 2020, IBM SPSS Statistics for Windows, Version 27.0, Armonk, NY, USA: IBM Corp).

## 3. Results

The study included 121 patients, with 39 (32.2%) women and 82 (67.8%) men, aged from 18 to 82. In each of the age groups from 18 to 40 years and from 41 to 60 years, there were 36 (29.8%) patients. In the group from 61 to 82 years, there were 49 (40.4%) patients. The patients were divided into four groups depending on the injury mechanism. The first group consisted of 55 (45.5%) patients with multi-organ injuries (over two injuries), the second consisted of 40 (33.0%) patients with isolated craniocerebral trauma, the third consisted of 25 (20.7%) patients after a surgery, and the fourth consisted of one person (0.8%) with spinal injury (Table 1).

In 59 (48.8%) of the studied cases, the symptoms of pneumonia developed within the 4th day after implementing mechanical ventilation, and in 62 (51.2%), pneumonia occurred over 4 days after starting mechanical ventilation.

### 3.1. Parameters of Nutritional Status

On the day of admission into the ward, their nutritional status was evaluated using the NRS-2002 scale. The whole group (121 persons) scored 3 or more points in the NRS-2002 scale, which points to the occurrence of under-nutrition risk and the necessity for nutritional treatment. In 45 (37.2%) patients, the total protein level in blood serum was at a level lower than 45 g/L, which is considered a critical value. In 75 (62.0%) cases, the value of total protein in serum was within the norm—from 45 g/L to 80 g/L. In one person (0.8%), the value was at a level higher than 80 g/L.

The measurement of albumin level in blood serum indicated correct values in 17 (14.0%) of the studied patients. In 45 (37.2%) cases, however, the values were at a level lower than 25 g/L, which indicates severe under-nutrition and a bad prognostic factor. In 32 (26.4%), the values showed moderate under-nutrition, and in 27 (22.4%), the values showed mild under-nutrition.

### 3.2. Inflammation Indicators

In 20 (16.5%) persons in the research group, the concentration of acute-phase protein (APP) occurred at a level lower than 5 mg/L; in 30 (24.8%) patients, the indicator was at a level from 6 to 40 mg/L; in 46 (38.0%) patients, from 41 to 200 mg/L; whereas in the remaining 25 (20.7%) patients, at a level from 200 to 500 mg/L.

The leukocytes level (WBC) in blood in 33 (27.3%) patients was within the norm, and in 5 (4.1%) patients, was below the physiological value. The most numerous group of studied patients—83 (68.6%)—presented a WBC indicator below the norm. The concentration of procalcitonin in blood higher than 0.5 µg/L was detected in 68 (56.2%) patients, and in 53 (43.8%), the values were normal.

### 3.3. The Evaluation of the Patients’ Condition Severity on Admission to the Ward

In order to assess the condition severity of patients admitted to the ICU, and to forecast their mortality risk, the APACHE II scale was used. Mortality risk ranged from 15–85% in the whole research group. Mortality risk from 15–40% occurred in 64 (50.8%) cases, and from 55–85% in 57 (47.5%) studied persons (Table 2).

### 3.4. Epidemiological Factors for VAP

With the aim of recognising the epidemiological factors responsible for the contraction of ventilator-associated pneumonia, the results of the inoculation obtained from the secretion from the patients’ respiratory tracts were subjected to analysis. The study demonstrated that pneumonia developed in the majority of the patients possibly because of non-fermenting Gram-negative bacilli *Acinetobacter baumannii* (*n* = 22), constituting 28.2% of all isolated bacteria, and Gram-positive strains *Staphylococcus aureus* (*n* = 22), constituting 28.2% of all grown bacteria. Detailed data are presented in Table 3.

From the obtained cultures, a positive increase was observed also in the range of the genus *Candida*. In the cultures, fungi of the species *Candida albicans* were predominant, but a significant species variation of the grown fungi was also detected. The presence of fungi from the genus *Candida* in the study has to be regarded as a colonization rather than an ethological factor of pneumoniae (Table 4).

### 3.5. Statistical Analysis of Selected Factors Determining the Occurrence of VAP

The analysis demonstrated a lack of correlation between the type of injury and the time of VAP contraction in the studied group. Despite the lack of statistically significant correlation, it was observed, however, that in patients after multi-organ injuries, VAP was diagnosed more frequently from the 5th day after intubation, whereas in patients with injuries related to the prior surgery, VAP was diagnosed more frequently up to 4 days after intubation (Table 5).

Also, the statistical analysis did not show any correlation between socio-demographic variables (age and gender) and nutritional status vs. contraction time of VAP in the studied group. Despite the lack of statistically significant correlation, it was observed that, in the case of an albumin value below 25 g/L, VAP developed more often (compared to the other values) up to 4 days after intubation (Table 6).

The statistical analysis demonstrated a significant difference in the development of ventilator-associated pneumonia in the studied patients depending on the level of leukocytes in the blood. It was proved that a higher level of leukocytes in the blood of patients with injury resulted in longer time of pneumonia contraction. No significant difference in the contraction of VAP in the group with various levels of acute phase protein and procalcitonin was detected. It was observed, however, that in the case of an acute phase protein value above 41 mg/L, VAP was diagnosed more often up to 4 days after starting intubation (Table 7).

The analysis demonstrated that there is no statistically significant difference between the contraction time of VAP in the group of studied patients with various numbers of points obtained in the APACHE II scale (Table 8).

The analysis of the correlation between epidemiological factors and VAP demonstrated that there is a statistically significant difference in contraction time of VAP in patients showing various epidemiological factors. In the case of VAP developing within 4 days, the most frequent epidemiological factors were *Staphylococcus aureus* and *Streptococcus pneumoniae*, whereas in the case of VAP diagnosed from 5th day, they were *Acinetobacter baumannii*, *Candida albicans*, and *Klebsiella pneumoniae* (*p* = 0.035) (Table 9).

In the further analysis, the dependency between the combination of various risk factors (the occurrence of various risk factors in one patient simultaneously) and VAP contraction time was determined. The analysis included 9 risk factors:albumin level in serum above 35 g/L;CRP value above 5 mg/L;leukocytes level above 10,000/µL;procalcitonin level above 0.5 µg/L;result ≥ 3 in the NRS-2002 scale;age over 61 years;gender—male;multi-organ injury;total protein level in serum below 45 g/L.

The statistical analysis included the group of patients in whom five or more risk factors were recognized due to the fact that, in this group, VAP occurred more often up to 4 days of intubation. The statistical analysis did not demonstrate any important influence of the number of risk factors in the studied patients on the time of pneumonia occurrence (Table 10).

In the final model aiming to assess the factors which influenced the appearance of the late VAP, two variables were retained: epidemiologic factor and APACHE II score. Concerning epidemiologic factors, patients infected with Staphylococcus aureus had 4% odds for the development of late VAP in comparison to Acinetobacter baumannii (reference category), whereas patients infected by any other bacteria or fungi than the two above mentioned ones had about four times lower odds of the development of late VAP in comparison to Acinetobacter baumannii. As concerns the score of the APACHE II scale, patients with results from 20 to 24 and from 25 to 29 had 13% and 21%, respectively, odds of the development of late VAP in comparison to patients with APACHE II scores ranging from 10 to 19 (Table 11).

## 4. Discussion

Ventilator-associated pneumonia (VAP) is a serious issue both clinically and epidemiologically for each ICU. It constitutes additional ailment to the already existing complications worsening the patient’s clinical condition, which poses a huge risk of death. Considering the above, special attention has to be paid to actions preventing VAP, and in early diagnosis and the implementation of effective treatment. In the authors’ own research conducted on a group of patients after experiencing injuries with diagnosed VAP, they were predominantly persons over 61 years of age (40.4%) and men (67.8%). The most frequent type of injury was multi-organ injury (45.5%).

The study by Walaszek et al., which included over 1200 patients mechanically ventilated, demonstrated that the largest VAP incidence occurred in the group with multi-organ injuries in the age range between 51 and 75 years [23]. The study by Duszyńska et al. on a group of 1097 patients demonstrated, in turn, that VAP developed mainly in males and patients after surgical procedures [37]. Other studies also confirm that an age of more than 60 years, male gender, and multi-organ injury are considered crucial risk factors for the contraction of VAP [38,39]. Slightly different results were obtained by Gianakis et al., who demonstrated that the highest risk of VAP contraction occurs in the post-trauma group at an average of 45 years [40].

One of important risk factors for VAP occurrence is abnormal nutritional conditions, especially under-nutrition. The entire study group obtained on the NRS-2002 scale a result which indicates the risk of under-nutrition and the necessity to implement nutritional treatment. The implementation of nutritional treatment is confirmed by ESPEN recommendations [41]. The results of our own research also exposed the fact that, after excluding patients with liver or renal failure features from the study, in 86% of patients, the parameter was below the norm threshold, with 37.2% below 25 g/L, and only 14% of the studied patients obtained the correct albumin level. Despite the more frequent remark that hypoalbuminemia ought not to be considered a result of under-nutrition, but rather as an indicator of the disease severity, inflammation, or hydration status of the organism, albumins are included in all prognostic-nutritional parameters, and their concentration in blood below 3.5 g/dL is pointed at as a factor of under-nutrition, and constitutes a bad probiotic factor, as well as serious risk for complications [42,43].

Of great importance for monitoring infections in ICUs are biomarkers: C-reactive protein (CRP) and procalcitonin (PCT) [44].

In our own research, 83.5% of patients obtained higher values of the CRP indicator, with the value of the parameter over 201 mg/L in 20.7% of cases, whereas in 68.6% of the studied patients, the level of leukocytes exceeded the norm (i.e., 10,000/µL), and 56.2% were indicated with a procalcitonin level over 0.5 µg/L.

Studies by other authors demonstrate that, in patients subjected to mechanical ventilation, the daily monitoring of CRP was useful for VAP prognosis, whereas PCT turned out to be a poor marker for VAP prognosis [45]. The study by Luyt et al. [46] demonstrated a positive prognostic value of PCT increase in blood serum in the case of pulmonary or extrapulmonary infections, which justified immediate antibiotic administration. The study conducted by Krzemińska et al. obtained a similarly high level of leukocytes, which indicated an ongoing severe infection, confirmed by the presence of the strain responsible for VAP. Also, the increased level of CRP on particular days confirmed the development of the inflammation process [38].

The statistical analysis performed in our own research demonstrated the correlation between the level of leukocytes on the day of admitting the patient into the ward and the day of VAP occurrence. The higher level of leukocytes was observed, the later pneumonia related to mechanical ventilation developed in the studied group.

The most frequent epidemiological factors for the development of early- and late-onset VAP in our own research were: *Acinetobacter baumannii* (18.2%), *Staphylococcus aureus* (18.2%), *Escherichia coli* (6.6%), and *Klebsiella pneumoniae* (5.8%). A correlation between the epidemiological factor and the day of VAP onset was also indicated in the research. An early onset of VAP was usually caused by bacteria existing in the oral cavity and throat, such as *Staphylococcus aureus* and *Streptococcus pneumoniae*, whereas late-onset VAP was usually caused by hospital pathogens, such as the strains of *Acinetobacter baumannii* and *Klebsiella pneumoniae*. Both epidemiological factors and the APACHE II scale scores proved to be important predicators of late VAP occurrence in our own research. Early detection of those factors may allow the implementation of prophylactic and therapeutic treatments, thus preventing possible complications.

In the study by Duszyńska et al., the most frequent epidemiological factor was *Acinetobacter baumannii* [37], whereas in the study by Adarsh et al., it was *Streptococcus* [39].

The overview of publications performed by Kharel and Bist in the area of South-East Asia demonstrated a variable frequency of VAP occurrence with increasing mortality. It was determined that VAP is a serious problem in ICUs, resulting in high costs and an appearing resistance to antibiotics. In order to potentially decrease future risk of VAP in this area, it is necessary to plan implementation of various intervention educational projects, such as: staff trainings, hygiene awareness, and constant monitoring, along with the implementation of antibiotics administration [47].

The results of our own research demonstrate a necessity to draw attention to different aspects which may influence the development of pneumonia related to mechanical ventilation. The knowledge of risk factors and mechanisms of VAP occurrence constitutes a chance to implement effective prophylactics.

## 5. Conclusions

Our study demonstrated that the development of pneumonia in mechanically ventilated patients (VAP) depends on numerous factors, including multi-organ injury, albumin and leucocytes levels, the CRP value, an age over 61 years, and the male gender. The time of VAP development, on the other hand, depended on the epidemiological factor (*Staphylococcus aureus*, *Streptococcus pneumoniae*—early VAP; and *Acinetobacter baumannii, Klebsiella pneumoniae*—late VAP), leucocytes level, and patients’ condition severity assessment according to the APACHE scale. Early implementation of nutritional treatment in patients with diagnosed VAP, and the knowledge of the above factors may prove to be useful for identifying patients with the risk of VAP, and the modification of medical care on patients to minimize the development of VAP.

## Figures and Tables

**Table 1 ijerph-19-00602-t001:** Type of injury recognized in a patient upon admission to ICU.

Type of Recognized Injury	*n*	%
Multi-organ injury (over two injuries)	55	45.5
Isolated craniocerebral injury	40	33
Post-surgery injury/trauma	25	20.7
Isolated spinal/pelvic injury	1	0.8

**Table 2 ijerph-19-00602-t002:** Evaluation of the patients’ condition severity on admission to the ward based on the APACHE II scale.

Number of Points	Mortality Risk	*n*	%
10–14 points	15%	8	6.6
15–19 points	25%	22	18.02
20–24 points	40%	34	28.0
25–29 points	55%	29	24.0
30–34 points	75%	25	20.7
>34 points	85%	3	2.5

**Table 3 ijerph-19-00602-t003:** Epidemiological factors of pneumonia in the studied group based on the inoculation from the respiratory tract secretion—bacteria.

Epidemiological Factor	*n*	%
*Acinetobacter baumannii*	22	28.2
*Staphylococcus aureus*	22	28.2
*Escherichia coli*	8	10.3
*Klebsiella pneumoniae*	7	9.0
*Streptococcus pneumoniae*	5	6.4
*Serratia marcescens*	2	2.5
*Stenotrophomonas maltophilia*	2	2.5
*Pseudomonas aeruginosa*	2	2.5
*Prevotella bivia*	1	1.3
*Enterobacter cloacae complex*	1	1.3
*Morganella morganii*	1	1.3
*Haemophilus influenzae*	1	1.3
*Klebsiella oxytoca*	1	1.3
*Streptococcus group C*	1	1.3
*Enterobacter cloacae*	1	1.3
*Moraxella catarrhalis*	1	1.3

**Table 4 ijerph-19-00602-t004:** Epidemiological factors of pneumonia in the studied group based on the inoculation from the respiratory tract secretion—yeast-like fungi.

Epidemiologic Factor	*n*	%
*Candida albicans*	34	79.1
*Candida dubliniensis*	2	4.7
*Candida tropicalis*	2	4.7
*Candida glabrata*	1	2.3
*Candida kefyr*	1	2.3
*Candida lusitaniae*	1	2.3
*Candida* sp.	1	2.3
*Candida krusei*	1	2.3

**Table 5 ijerph-19-00602-t005:** The correlation between the type of injury and VAP contraction time.

Type of Injury	Up to 4 Days VAP		From 5th Day VAP		Chi^2^	*p*
	*n*	%	*n*	%		
multi-organ (>2 injuries)	23	37.1	32	54.2	6.04	0.11
craniocerebral	22	35.5	18	30.5		
surgery-related	17	27.4	8	13.6		
spinal/pelvic	0	0.0	1	1.7		

*p* value for the chi^2^ test—level of statistical significance.

**Table 6 ijerph-19-00602-t006:** The correlation between selected parameters of nutritional status and VAP contraction time.

Variable	Up to 4 Days VAP		From 5th Day VAP		Chi^2^	*p*
	*n*	%	*n*	%		
Total Protein Level						
<45 g/L	22	35.5	23	39	1.07	0.586
45–80 g/L	39	62.9	36	61		
>80 g/L	1	1.6	0	0		
Albumin level in blood serum						
<25 g/L	27	43.5	18	30.5	2.95	0.4
25–30.99 g/L	13	21	19	32.2		
31–35 g/L	13	21	14	23.7		
>35 g/L	9	14.5	8	13.6		

*p* value for the chi^2^ test—level of statistical significance.

**Table 7 ijerph-19-00602-t007:** The correlation between selected inflammation indicators and VAP contraction time.

Variable	Up to 4 Days VAP		From 5th Day VAP		Chi^2^	*p*
	n	%	n	%		
Acute Phase Protein Level						
<5 mg/L	10	16.1	10	17	3.81	0.282
6–40 mg/L	11	17.8	19	32.2		
41–200 mg/L	27	43.5	19	32.2		
201–500 mg/L	14	22.6	11	18.6		
Leukocytes level						
<4000/µL	5	8.1	0	0	8.84	0.012 *
4000–10,000/µL	21	33.9	12	20.3		
>10,000/µL	36	58	47	79.7		
Procalcitonin level						
<0.5 µg/L	27	43.5	26	44.1	0.003	0.954
>0.5 µg/L	35	56.5	33	55.9		

* *p* value for the chi^2^ test—statistically significant.

**Table 8 ijerph-19-00602-t008:** The correlation between the results of the APACHE II scale and VAP contraction time.

APACHE II Scale Level	Up to 4 Days VAP		From 5th Day VAP		Chi^2^	*p*
	n	%	n	%		
10–14 points	4	6.5	4	6.8	6.026	0.304
15–19 points	9	14.5	13	22		
20–24 points	20	32.3	14	23.7		
25–29 points	16	25.8	13	22		
30–34 points	10	16.1	15	25.4		
>34 points	3	4.8	0	0		

*p* value for the chi^2^ test—level of statistical significance.

**Table 9 ijerph-19-00602-t009:** The correlation between epidemiological factor VAP contraction time.

Epidemiological Factor	Up to 4 Days VAP		From 5th Day VAP		Chi^2^	*p*
	n	%	n	%		
*Staphylococcus aureus*	18	29	4	6.8	37.91	0.035 *
*Acinetobacter baumannii*	6	9.7	16	27.1		
*Escherichia coli*	5	8.1	3	5.1		
*Streptococcus pneumoniae*	4	6.5	1	1.7		
*Klebsiella pneumoniae*	3	4.8	4	6.8		
*Pseudomonas aeruginosa*	2	3.2	0	0		
*Serratia marcescens*	2	3.2	0	0		
*Haemophilus influenzae*	1	1.6	0	0		
*Klebsiella oxytoca*	1	1.6	0	0		
*Moraxella catarrhalis*	1	1.6	0	0		
*Morganella morganii*	1	1.6	0	0		
*Prevotella bivia*	1	1.6	0	0		
*Stenotrophomonas maltophilia*	1	1.6	0	0		
*Enterobacter cloacae*	0	0	1	1.7		
*Enterobacter cloacae complex*	0	0	1	1.7		
*Stenotrophomonas maltophilia*	0	0	1	1.7		
*Streptococcus group C*	0	0	1	1.7		

* *p* value for the chi^2^ test—statistically significant.

**Table 10 ijerph-19-00602-t010:** The influence of the risk factor number in studied patients on VAP occurrence time.

Variable	Day of Occurrence of Pneumonia	MEAN	U Test	*p*
Number of risk factors	Up to 4 days VAP	6	1729	0.598
	From 5th day VAP	6		

MEAN—arithmetical mean value; U test—non-parametric Mann–Whitney test; *p* value—level of statistical significance.

**Table 11 ijerph-19-00602-t011:** The correlation between VAP epidemiological factors and APACHE II scores, and the time of late VAP occurrence.

Without Candida Albicans *N* = 86	OR	95% CI		*p*
Epidemiological factor based on inoculation from respiratory tract excretion				
Acinetobacter baumannii	1			
Staphylococcus aureus	0.04	0.01	0.22	<0.001 *
Other	0.24	0.07	0.80	0.020 *
APACHE II scores				
10–19 (25%)	1			
20–24 (40%)	0.13	0.03	0.56	0.006 *
25–29 (55%)	0.21	0.05	0.85	0.028 *
>30 (75%)	0.72	0.15	3.43	0.681

OR—odds ratio, CI—95% confidence intervals, * *p* value—level of statistical significance.

## Data Availability

The data presented in this study are available on reasonable request from the corresponding author.

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
