# Peer review of "Nutritional Status Disorders and Selected Risk Factors of Ventilator-Associated Pneumonia (VAP) in Patients Treated in the Intensive Care Ward—A Retrospective Study"

_ijerph, 2022, doi:10.3390/ijerph19010602_

Round 1

Reviewer 1 Report

These are the observation to be addressed

  1. How to measure the nutritional status of the patients in ICU.
  2. In the sedation state of a patient, how much % of the nutrition decrease in the patient's body.
  3.  In results section 3. in the sample profile, the authors mentioned total participants was 121, four groups (Group 1: 50, Group 2: 40, Group 3: 25, and Group 4: 1) total participants (50+40+25+1+116).  Here confusion arises. 
  4. The hypothesis is not clear in the study.
  5. What is the significance level (95% or 99%) nowhere mentioned?
  6. In this study, only chi Square test – level of statistical significance was used.
  7. The ANOVA approach to be applied or  any Univariate analysis 

Author Response

Dear Reviewer,

Thank you very much for all valuable comments and suggestions. We have made corrections, checked all text and change it according to reviewer comments.

1. How to measure the nutritional status of the patients in ICU.

There are no validated and recommended tools for assessment of nutritional status in a critically ill patient. Various scales are recommended to be used in screening tests on patients admitted to intensive care units, namely Nutrition Risk Score (NRS 2002), NUTRIC (Nutrition Risk in the Critically Ill) Score, Subjective Global Assessment (SGA) or Malnutrition Universal Screening Tool (MUST) [Weimann, A.; Braga, M.; Carli, F.; Higashiguchi, T.; Hubner, M.; Klek, S.; Laviano, A.; Ljungqvist, O.; Lobo, D.N.; Martindale, R.G.; et al. ESPEN practical guideline: Clinical nutrition in surgery. Clin. Nutr. 2021, 40, 4745–4761].  In own research, to assess nutritional status of an adult patient admitted to ICU the NRS 2002 scale was used, according to the recommendation of Polish Ministry of Health (Minister of Health regulation of 1st January 2012 as amended).

2.In the sedation state of a patient, how much % of the nutrition decrease in the patient's body.

We are sincerely grateful for the precious remarks and pointing to the essential feature of a patient's nutrition in the state of sedation. It has encouraged us to analyse the most recent scientific reports.  Sedation has become an integral part of the intensive care practice in mechanically ventilated patients. The most commonly used intravenous sedative for patients infected from a respirator is propofol. The usage of propofol for sedation helps to a great extent satisfy the patient's caloric demand. Due to the presence of oil carrier solution in propofol, intravenous infusion contains 1.1 kcal/ml. Propofol combined with enteral and parenteral nutrition constituted from 5% to 24% of the total calorie consumption. As a result, propofol may significantly contribute to the amount of consumed calories and may potentially cause complications of overfeeding in patients who are simultaneously receiving enteral and parenteral nutritional therapy. I order to avoid the potential over-nutrition, it is recommended to reduce the speed of infusion of nutritional treatment, which may also pose adverse effects because the consumption of protein may be insufficient [Dickerson, R. N., Buckley, C. T.: Impact of Propofol Sedation upon Caloric Overfeeding and Protein Inadequacy in Critically Ill Patients Receiving Nutrition Support. Pharmacy (Basel, Switzerland) 2021, 9(3), 121].

Individual caloric demand may be assessed by simultaneous measurement of  oxygen consumed and oxygen produced, the so-called indirect calorimetry. It was demonstrated that individual calculation of casloric demand using indirect calorimetry reduces complications and improves parenteral nutrition administration [Singer P, Anbar R, Cohen J et al. The tight calorie control study (TICACOS): a prospective, randomized, controlled pilot study of nutritional support in critically ill patients. Intensive Care Med 2011; 37: 601–9]. Both European and American guidelines recommend the usage of indirect calorimetry.

Increase of sedation time is considered a risk factor for development of ventilator-associated pneumonia (VAP). [Shahabi, M., Yousefi, H., Yazdannik, A. R., Alikiaii, B.  The effect of daily sedation interruption protocol on early incidence of ventilator-associated pneumonia among patients hospitalized in critical care units receiving mechanical ventilation. Iranian journal of nursing and midwifery research, 2016;21(5), 541–546.]

  1. In results section 3. in the sample profile, the authors mentioned total participants was 121, four groups (Group 1: 50, Group 2: 40, Group 3: 25, and Group 4: 1) total participants (50+40+25+1+116).  Here confusion arises

Thank you for this assessment.  An error appeared in the text of the article: the first group listed 55 patients multi-organ injury, according to the data from Table No. 1, the number of patients is correct - 55.

  1. The hypothesis is not clear in the study.

Thank you for your valuable suggestion. The study assumed the following hypothesis: The more risk factors, the higher probability of development of ventilation-related pneumonia.

  1. What is the significance level (95% or 99%) nowhere mentioned?

The level of significance has been added – 95%. We add it in the text.

  1. The ANOVA approach to be applied or  any Univariate analysis 

The analysis has been repeated according to the suggestions. Additional analysis of regression has been performed, according to the reviewer's suggestion.

Reviewer 2 Report

The authors present a retrospective trial on VAP in a single center MICU population. They find a correlation between admission diagnosis , nutritional status , inflammatory markers, severity of illness and age to VAP development. 

The trial has some flaws that need to be addressed. In the current form it should not be published.

  1. I do not fully understand the patient population. Baseline data for basic things like age and gender distribution and morbidity markers as IMV duration, ICU LOS, HOS LOS, and mortality are not clear. This needs to be addressed. I do not understand the setting - are these trauma patients ? The word multi organ injury is unclear. Does this address trauma or do the authors mean dysfunction? If they are trauma patients why are they on the MICU?
  2. The main focus of this trial is as I understand to examine nutritional status and development of VAP. The authors use a risk assessment on admission. All patients are at risk. So how is this helpful? How do you adress this risk in an ICU setting? Is optimisation of nutrition in initial or late phases of ICU treatment a factor ? What is their nutritional practice ? 
  3. I don't understand the paragraph about "correct" values. Please explain. What does correct total protein 45-80g/dl mean ? What are incorrect values in this setting? Is the undernutrition just bases in albumin / protein ? This needs to be explained and would be worrisome as low albumin can be a result of a plethora of things especially - aside from nutritional status - due to sepsis, volume resuscitation, capillary leak, abdominal surgery, ......
  4. Analysis: all analysis is univariate? Is there a multivariate analysis I am missing to see?  Please explain / address.
  5. The authors state that they see many patients with late VAP due to Candida sp. This is of course complete nonsense. Candida does not cause pneumonia but is a (mal-)colonisation of the airways without any indication for treatment. So the complete result section if irrelevant in this regard.
  6. APACHE scores: The stated scores are not helpful in interpreting the data given the lack of other outcome data. 
  7. Formating: The text is too long for the items addressed. The tables are not easy enough to grasp. This should be changed once the other factors have been resolved.

Author Response

Dear Reviewer,

thank you very much for all valuable comments and suggestions. We have made corrections, checked all text and change it according to reviewer comments.

  1. I do not fully understand the patient population. Baseline data for basic things like age and gender distribution and morbidity markers as IMV duration, ICU LOS, HOS LOS, and mortality are not clear. This needs to be addressed. I do not understand the setting - are these trauma patients? The word multi organ injury is unclear. Does this address trauma or do the authors mean dysfunction? If they are trauma patients why are they on the MICU?

The study included patients treated in ICU with recognised injuries, with diagnosed ventilation-related pneumonia which developed after at least 48 hours of intubation. The exclusion criteria from the study were mechanically ventilated patients in another ward, with diagnosed pneumonia at the time of admission into ICU, as well as patients with symptoms of liver or renal failure. In the study, the group of patients with multi-organ injury consisted of persons who had been in an accident/injured causing major injuries of at least two organs, which posed a risk of impaired cardiopulmonary stability in a patient, and each of those injuries required a specialistic treatment in the ICU due to the life-threatening condition.

  1. The main focus of this trial is as I understand to examine nutritional status and development of VAP. The authors use a risk assessment on admission. All patients are at risk. So how is this helpful? How do you adress this risk in an ICU setting? Is optimisation of nutrition in initial or late phases of ICU treatment a factor? What is their nutritional practice? 

The main goal of the study was to determine the correlation between selected risk factors (nutritional status at the time of admission into the ICU, type of injury, epidemiological factors, risk of death, inflammation parameters, age and gender) and the time of pneumonia occurrence in patients with injuries who were mechanically ventilated in intensive care units. The assessment of nutritional status was one of risk factors of VAP occurrence. The authors' goal was not only to assess nutritional status. According to the accepted inclusion criteria for the study, the analysis was performed on the data of mechanically ventilated patients in whom pneumonia symptoms developed after at least 48 hours from intubation. In all patients at the time of admission the risk of malnutrition was determined based on the NRS 2002 scale. According to the ESPEN recommendations, due to the patients' clinical condition, taking also into consideration the results of nutritional status assessment results in the NRS 2002 scale, parenteral nutrition was introduced. Nutrition content is prepared according to the clinical condition of patients and their metabolic needs. In the clinical ward where the study was performed, in order to determine energetic demands a formula was implemented according to which the aim of nutritional treatment was to provide between 20 and 25 kcal/kg/day for the patient. Non-protein energy from carbohydrates and fats constituted 50-65% of energetic demand. The supply of protein was 1.5-2.5 g/kg/day, and demand for water calculated per kilogram of body mass was 30-40 ml.].  In the own studies, to assess the nutritional status of an adult patient admitted into the ICU the NRS 2002 scale was used, which is recommended by the Polish Ministry of Health (Minister of Health regulation of 1st January 2012 as amended).

  1. I don't understand the paragraph about "correct" values. Please explain. What does correct total protein 45-80g/dl mean? What are incorrect values in this setting? Is the undernutrition just bases in albumin / protein? This needs to be explained and would be worrisome as low albumin can be a result of a plethora of things especially - aside from nutritional status - due to sepsis, volume resuscitation, capillary leak, abdominal surgery, ......

For the purpose of the study, reference values of total protein in blood serum were accepted at 45-80 g/dl. Values below 45 g/dl are incorrect according to the reference norms of the laboratory performing the tests University Hospital in Krakow. We agree that the low albumin level may occur in different clinical conditions, however, in our research in the case of patients treated in the ICU the albumin level was accepted as the parameter of initial assessment of nutritional status, and patients with symptoms of liver or renal failure were excluded. The choice of these parameters was guided by the results of other authors' studies. They demonstrate that biochemical studies covering the level of albumin and protein concentration are useful in recognition of protein-caloric under-nutrition. Although it is more and more frequently noted that hypoalbuminemia should not be considered a result of under-nutrition, but first of all as an indicator of the disease severity, inflammation or the level of organism hydration, albumins are included in the prognostic-nutritional parameters, and their concentration in blood below 3,5 g/dl is a listed malnutrition measurement and constitutes a negative prognostic factor, as well as it indicates severe risk of complications [McClave, S.A., Taylor, B.E., Martindale, R.G. et al; Society of Critical Care Medicine American Society for Parenteral and Enteral Nutrition. Guidelines for the Provision and Assessment of Nutrition Support Therapy in the Adult Critically Ill Patient: Society of Critical Care Medicine (SCCM) and American Society for Parenteral and Enteral Nutrition (A.S.P.E.N.). JPEN J Parenter Enteral Nutr 2016 Feb, 40(2), 159-211; SzczygieÅ‚, B. Niedożywienie zwiÄ…zane z chorobÄ…, PZWL: Warszawa, Poland, 2017; pp. 9-22.

  1. Thank you very much for your valuable advice. Analysis: all analysis is univariate? Is there a multivariate analysis I am missing to see?  Please explain / address.

Thank you very much for your valuable advice. It was performed according to the multi-dimensional analysis recommendations.

  1. The authors state that they see many patients with late VAP due to Candida sp. This is of course complete nonsense. Candida does not cause pneumonia but is a (mal-)colonisation of the airways without any indication for treatment. So the complete result section if irrelevant in this regard.

According to the suggestion, fragments of the text on Candid have been removed from the section of Results, Discussion and Conclusions.

  1. APACHE scores: The stated scores are not helpful in interpreting the data given the lack of other outcome data. 

The content related to the APACHE scale has been used in regression model of analysis.

  1. Formating: The text is too long for the items addressed. The tables are not easy enough to grasp. This should be changed once the other factors have been resolved.

The text volume has been reduced. The tables have been corrected.

Round 2

Reviewer 2 Report

The authors have addressed all concerns of the previous versions and have markedly improved the overall appearance, clearness and significance of the manuscript. Except for minor spelling checks I fully support publication of this manuscript as presented now. Good work.